# Human Hand Anatomy-Based Prosthetic Hand

**DOI:** 10.3390/s21010137

**Published:** 2020-12-28

**Authors:** Larisa Dunai, Martin Novak, Carmen García Espert

**Affiliations:** 1Centro de Investigación en Tecnologías Gráficas, Universitat Politècnica de València, camino de Vera s/n, 46022 Valencia, Spain; 2Faculty of Mechanical Engineering, Czech Technical University in Prague, Technická 4, Praha 6, 166 00 Prague, Czech Republic; Martin.Novak@fs.cvut.cz; 3Hospital La Fe, Avinguda de Fernando Abril Martorell, 106, 46026 Valencia, Spain; garcia_caresp@gva.es

**Keywords:** prosthetic hand, MyWare sensor, force sensing resistors, human hand anatomy

## Abstract

The present paper describes the development of a prosthetic hand based on human hand anatomy. The hand phalanges are printed with 3D printing with Polylactic Acid material. One of the main contributions is the investigation on the prosthetic hand joins; the proposed design enables one to create personalized joins that provide the prosthetic hand a high level of movement by increasing the degrees of freedom of the fingers. Moreover, the driven wire tendons show a progressive grasping movement, being the friction of the tendons with the phalanges very low. Another important point is the use of force sensitive resistors (FSR) for simulating the hand touch pressure. These are used for the grasping stop simulating touch pressure of the fingers. Surface Electromyogram (EMG) sensors allow the user to control the prosthetic hand-grasping start. Their use may provide the prosthetic hand the possibility of the classification of the hand movements. The practical results included in the paper prove the importance of the soft joins for the object manipulation and to get adapted to the object surface. Finally, the force sensitive sensors allow the prosthesis to actuate more naturally by adding conditions and classifications to the Electromyogram sensor.

## 1. Introduction

More than 3 million people suffer from hand amputations or loss due to health disorders caused by infections, congenital absence, diabetes, cancer or others [1,2]. Over 75% of the amputations are partial [3]. Hand loss has an important impact on the person’s functional aspect. Many of the people with a loss of the hand have the possibility of using a prosthetic hand. The development of prosthetic hands has been less based on their functionality, relying more on human hand aesthetic aspects [4,5,6,7,8]. With the technological advances in biotechnology, the innovation reached the area of robotics and prosthetic hand development. Consequently, current commercial prosthetic hands have become more sophisticated. They are fitted with sensors and actuators, so that the fingers are motorized and can perform grasping movements. Nevertheless, automatized prosthetic hands are expensive and not accessible to all social strata. Usually, the most common prosthetic hands are passive, and their goal is to substitute the human hand more esthetically than functionally. Powered prosthetic hands are classified in body powered and external powered prosthetic hands [9]. Body powered prosthetic hand mechanisms are actuated by human body movement through wires or cables. Usually, these types of devices are simple devices with grasping movement and are relatively lightweight. Moreover, body powered prosthetic hands require harnessing. External powered prosthetic hands are based on external power and actuators. Some of these types of prosthetic hands are controlled by Electromyograms (EMGs) [10,11] for grasping. The most common EMG-controlled prosthetic hands use surface EMG [12] while a few others use intramuscular EMG [13,14]. EMG prosthetic hands are amplitude-based measurement devices and, usually, the control is slow. Because most of the prosthetic hands are controlled by a single input, the control of individual fingers or joins is not allowed. Usually, the prosthetic hands based on EMG use electrical signals of two antagonist muscle contractions. They allow two directions of movement: flexion and extension; one is for start grasping while the other is to start extension. As the EMG based prosthetic hands do not have external cables, these devices are more esthetical. To obtain more than two movements for the prosthetic hand, it is required to introduce more conditions, such as triggering or artificial intelligence (pattern recognition and classification). Prosthetic hands that are aimed to perform movements for all fingers operate in a sequential order with time delay. In some prosthetic hands, the movements of the different fingers are performed by using several contractions of the same muscle (quick contractions of the same muscle) or by alternating both muscle contractions to control different joint movements. Another control system is based on force-sensing resistors, pull or push switches or Inertial Measurement Unit (IMU) [15].

Prosthetic hands also include hybrid prostheses. Hybrid prostheses are body powered and externally powered devices. Often, these devices are used in cases of upper limb amputations, including transhumeral and shoulder. Regarding external powered prosthesis devices, these can be classified as those with one degree of freedom and those with multiple degrees of freedom. Devices with one degree of freedom perform only the extension and flexion movements. Usually these devices are robust [16,17]. The ones with multiple degrees of freedom, also known as multiarticulated prostheses, are fitted with several actuators for different fingers and/or interphalangeal joins [17,18]. They use small actuators that perform the required movement. Despite the high accuracy of the EMG signals, the researchers are still looking for the best methods of prosthetic hand control by combining EMG with artificial vision [19], microphone [20], tongue control system [21], etc. Table 1 presents the characteristics of the most advanced bionic prosthetic hands.

The present study introduces novel prosthetic hand development, based on a combined control system that uses EMG, buttons, and force-sensing resistors. The device design is totally based on the human hand anatomy. All phalanges are human hand-scanned phalanges. The ligaments and joins are strictly developed as real ones. The device has 15 DOF (Degree of Freedom) and the joins have different speeds and forces. The soft material joins provide the prosthetic hand with a high level of adaptation to the object surface. They increase the DOF of each join, enabling the small abduction/adduction. The use of force sensitive resistors allows the prosthetic hand to simulate the touch pressure sensing that stops grasping movement. 

The paper is structured as follows: Section 2 describes the materials and methods of the prosthetic hand development. Section 3 presents the experimental results and, finally, in Section 4, the conclusions are provided.

## 2. Materials and Methods

### 2.1. Prototype Design

The prototype was built using a human hand anatomy-based design. All the elements of the prosthetic hand were based on real human hand measurements that included the dimensions, proportions, and human hand functionality. The idea of the proposed prosthesis relied on the reproduction of the human finger motions. For the prosthetic phalanges design, the real human hand phalanges were 3D scanned and then designed by using a 3D drawing technology—the Autodesk Inventor Professional 2019. The whole prosthetic hand structure for actuators and processing supports was modelled with the same 3D drawing tool. Before proceeding to the prosthetic hand design and assembly, the main design specifications based on the human hand behavior as joins and movement capabilities were analyzed. All hard elements were constructed by using 3D printing technology with Polylactic Acid (PLA) filament that has good functional and structural characteristics and that are suitable for 3D printing. One of the main novelties of this prototype relies on the employed materials, which are ideal to reproduce human tendons, ligaments, fibrous sheaths, joins, etc.

The human hand consists of Carpal bones, Metacarpal bones, Proximal, Middle and Distal Phalanges. All fingers are based on four bones: Metacarpal bone, Proximal, Middle and Distal Phalanges (see Figure 1). The thumb finger is different and has one phalange less than the rest of the fingers: Metacarpal bone and the Proximal and Distal phalanges. The joins are located between phalanges. There are fourteen joins for the whole hand. The join between Carpal and Metacarpal bones does not have any Degree of Freedom (DOF). The Thumb is the only one with a Metacarpal join with abduction/adduction movement with respect to the sagittal plane. The rest of joins have one DOF, flexion and extension movement with respect to the frontal plane. 

Taking into consideration the number of ligaments and their characteristics, the artificial ligaments are chosen from rubber materials with different hardness and elasticity characteristics.

It is known that the bone dimensions are important for the prosthetic hand design and development. The phalanges and Metacarpal bone lengths considered for the prosthetic hand are those corresponding to an adult female.

The average lengths of the human hand are presented in Table 2. The lengths of the phalanges significantly affect the object manipulation and hand movement. As the prosthetic phalanges are based on human hand anatomy, the length of the fingers is 99% of the real hand; the 1% remaining depends on the joins. As the joins are reproductions of the human hand joins, the abduction/adduction and rotation for each join is possible.

The abduction/adduction as well as the flexion/extension of the thumb are independently controlled by the control system. The kinematics of the Index finger are represented in Figure 2. The five fingers are driven by six actuators; each finger is controlled by one actuator, except on the Thumb, which is actuated by two. The purpose of this architecture, relying on using at least one actuator per finger, is to allow the prosthetic hand to perform finger movements independently.

As can be seen in Figure 2, the prosthetic hand kinematics is based on real-human hand anatomy. The solution for the joins is to design the volar plate, collateral ligaments, and extensor ligaments, as shown in Figure 1b,c using rubber materials with different hardness. The developed join elements are presented in Figure 3. The joins can perform 2 DOF at each join that allows them to increase their functionality. Nevertheless, the abduction/adduction movement of the phalanges is so small that it does not have the sense to introduce it in the prosthetic hand. The only existing abduction/adduction and flexion/extension movement in the four fingers is between the Metacarpal bone and the Proximal Phalange, in the MCP join.

To reduce the movement range, the role of the stopper is played by the tendon rail. All fingers are actuated through wires (tendons), which substitute flexor and extensor tendons connected to the actuators pulley. The mechanism for finger movement is based on the endless routing tendons, in which the flexor and extensor tendons are connected to the same actuator pulley (see Figure 4). This architecture enables driving the pulley in both directions at the same time. The assembly of the prosthetic hand is presented in Figure 5. The difference versus another devices is that, in this prosthetic model, the inter phalanges-driven pulley is not used. The tendon passes through tendons rails of each phalange and ends on the distal phalange. To avoid tendons tearing, an additional 15% of tendon is added to each finger.

For the hand control, five force sensing resistors are used. The sensors are placed on the Distal Phalange muscle and are built using soft flex material. The Artificial Abductor Muscle is fabricated with rubber and prevents the movement more than it is necessary. It also enables to complete the palm of the prosthesis. As the thumb join with the trapezoidal carpal bone is made by tendons and collateral ligaments, it allows the join to perform 3DOF (flexion/extension, abduction/adduction and turn).

The maximum motion angle of the thumb abduction is 80 degrees. For the other four fingers (the Index, Middle, Ring and the Little fingers), the motion flexion/extension angle is from 0 degrees to 90 degrees. Each join of the phalanges is dotted with artificial cartilages to avoid phalanges friction. The tendon sheath covers and holds the flexor and extensor tendons rails.

The actuators and EMG are placed in the prosthesis forearm, while the servomotor for the abduction/adduction movement of the thumb is located in the carpal bones of the prosthesis.

### 2.2. Control System

The electronic system of the prosthetic hand is based on a Myware EMG sensor, five step motors, five drivers, a servo motor, an Arduino AtMega 2560, two pushbuttons, a shield and five force sensing resistors. The characteristics of the electronic components are presented in the Table 3.

The main control unit of the prosthetic hand is the Arduino AtMega 2560. Prosthetic hand movement activation is controlled by a surface Myoware muscle sensor. The sensor electrodes are placed on the human hand skin above the flexor muscle and the reference electrode is installed in a neutral place (over the join bones), as shown in Figure 6.

The block diagram of the electronic circuit is shown in Figure 7. It provides the control signals STEP and DIR to all 5 stepper motors. A DRV8825 stepper motor driver is used. Each motor has its individual driver. The driver sets the motor current to the desired value to control the torque. To assure a more precise positioning, 1/32 microstepping is enabled. The used stepper motor type is 28BYJ-48 with an integrated reduction gearbox. The stepper motors operate the pulleys, as shown in Figure 4. The motion of the thumb is done with a servo, type Microservo SG90, connected directly to the Arduino. The force sensors, button and EMG sensor are connected to the Arduino as well. The FSR relates to a voltage divider to an analog input of the Arduino.

The control algorithm operates as a state machine. It starts in an IDLE state, where all stepper motors and the servo are stopped. In this state, the start command is awaited. It is assumed that the initial position of the hand is kept the same through the flexibility of the rubber tendons. The start signal can come from the button or the EMG sensor. When the start command is issued, the movement starts. The stepper motors and the servos are controlled independently to allow for the grasping of objects with different shapes. The grasping is completed when the FSR in the thumb signal is above a given threshold. Position (number of steps) for all steppers is stored in memory and used for the release phase.

Once the extension is required, the user presses the button for the extension and the hand automatically passes to the initial position and waits for the new command.

The signals received by EMG sensors are amplified and rectified. Figure 8a illustrates the EMG signal and in Figure 8b the filtered signal. The input amplitude signal of the EMG is in millivolts. Once the EMG signal is received, it should be filtered with a band-pass filter or by using a low-pass filter and a high-pass filter. Firstly, the signal is filtered with a digital Low-Pass Filter based on the Kirchoff’s Law in order to reduce signal noises. The amplification is calculated by:(1)x[n]=α∗y[n]+(1−α)∗y[n−1]
where α is the smoothing factor that varies from 0 to 1, x[n] is the resulting filtered discrete signal, y[n] is the discrete signal received by EMG. An example of an EMG filtered signal is presented in Figure 6b. In that figure, the EMG signal was filtered at α = 0.05. The EMG signal was tested for different α values from 0.05 to 1. For the α = 0.05 value, the filter is slower and clear. In Figure 8b the filtered signal is represented in red color.

The filtered signal is then classified and analyzed for finger flexion and extension. The features are extracted and discriminated from the EMG signal. The power for grasp movement is considered in the present study. The posture of the fingers is not considered.

In order to quickly calculate the EMG signal, the features are extracted in the time domain. Furthermore, the grasping function starts with the condition of EMG signal amplitude. It starts grasping when the EMG signal amplitude exceeds the predefined threshold.
(2)$$f\left(x\right)=\{\begin{array}{c} 1\;if\;x>threshold\\ 0\;otherwise \end{array}$$
where *f*(*x*) is the EMG input signal. The threshold is defined in accordance with the EMG signal voltage.

Figure 8 presents, in the red rectangle, the grasping movement. The signal below the amplitude threshold is not considered.

The grasping end depends on many factors as contact points, force closure, grasp control, external force, friction, etc. it means that it is necessary to realize object surface exploration.

Let us consider that external force is defined as f and depends on the external wrench *w*, at a moment *m*, the contact force *p* and torque *τ*.

The torque can be defined as:(3)$$\tau =p*J^{T}$$
where $J^{T}$ is the Jacobian matrix for manipulation. 

The force balance *f* = −G * w can be calculated by taking into consideration that the applied contact force must balance the external force applied to the object, where G is the grasp matrix.

The grasping force is also proportional to the actuators current *f* = k * I. If considering that the force-sensitive resistance sensor voltage *v*, then the contact force *p* = k * *v*.

The grasping stop function then is calculated as:(4)$$v=\{\begin{array}{c} 1\;if\;p>threshold\\ 0\;otherwise \end{array}$$

## 3. Results

The experiments carried out with the prosthetic hand aimed to verify the correct functionality of the device. Moreover, in these experiments, the evaluation of the device structure was also performed. To this end, the joins equilibrium was studied. As the joins are made of rubber, it is important to analyze the correct fingers flexion/extension trajectory, as well as the limits of the possible deviations of the phalanges under pressure. During the experiments, the prosthetic hand was placed in a vertical position fix-mounted to the table. The experimental study was based on the same manipulation motion as described in [27]. The main target was to establish the grasp movement. In the experiments, the same force sensing resistors threshold is used for all objects. That means that the prosthetic hand applied the same force to all objects. The grasp experiments were conducted with three different size objects; the objects employed in the experiments are: a ball, a pencil, and a note block, as shown in Figure 9. The ball diameter was 64 mm, the note block width was 4 mm and the pencil diameter was 10 mm. During the experiments, the movement of the fingers was observed. This movement allows us to better perceive the flexion, extension, abduction/adduction as well as the rotation for adaptation to the objects surface. The fingers trajectory of the hand for a simple grasping was also studied to analyze the small abductions/adductions on the fingers joins represented in Figure 9a. Generally, the trajectory defined by the prosthetic hand is cylindric pointing to the carpal bones. In the grasping of the tennis ball, shown in the Figure 9b, the ball is held into the hand. The grasping stops when the force sensing resistor detects the threshold. Then, it stops.

The hand can grasp the ball without any glove, but the use of a glove is also tested. The glove used in the experiments was a standard glove made of latex. Because the glove has a dry surface, it was impossible to grasp the tennis ball since it slipped from the hand. Additionally, the glove design influences the prosthetic hand grasping experiments. Afterwards, the ball grasping is tested without the latex glove, and the prosthetic hand can grasp the ball, as observed in Figure 9. Figure 9b shows that the prosthetic hand finger trajectory has been slightly displaced, adapting to the object surface, except on the thumb that performs a greater displacement. For the pen grasping, a clamp movement is perceived. The thumb applies the force to the pencil and the index Distal Phalange. When the force threshold is detected, the prosthetic hand stops grasping. For the notebook grasp, only the index and thumb fingers are operative, as shown in Figure 9d. In this case, the experiment consists in controlling the thickness and correct manipulation of thin objects. Figure 9a,b presents the abduction/adduction of the index and little finger. The maximum abduction of the Index finger is over 30° and 20° for the little finger. This deviation depends on the object size. The soft joins allow the fingers to adapt to the objects surface in order to obtain a better grasp.

The grasping time is different and depends on the object’s thickness or diameter. The basic grasping time was 1.3 s from the open hand position.

For a better object manipulation, independent phalange tendons that can be controlled by the same actuator incrementing joins kinematics and mechanics are necessary.

With regard to the design of the prosthetic hand, the rubber materials resistance was studied. The joins and the rubber-made extensor tendon, as well as the artificial adductor policis (muscle), are analyzed. The hand was evaluated and tested during 6 months in laboratory conditions (without patients) to test its functionalities and materials. During this period, multiple grasping movement experiments were performed. Some wear in the extensor tendon and artificial adductor muscle was observed after this period. The rubber started cracking and changed color in the joins zone, as well as where other rubber material was, as can be seen in the Figure 10. In addition, after 6 months, the artificial adductor muscle, made by the same material, broke when big objects were tested, such as the tennis ball. The rubber extensor tendon generated resistance when the grasping was performed, and this resistance required high-powered actuators. To solve this disadvantage, an additional extensor tendon was attached to the first, by using wires passing through the tendon rail. Small servomotors with the torque of 1.8 kg/cm were not enough. Due to this problem, the usage of the stepper motors was adopted in the current prosthetic hand. The weight, size and power supply of the prosthetic hand depended on the electronics and the prosthetic hand functionality. Finger load is correlated with the rate torque of the actuators.

One of the objectives of the presented prototype was to improve prosthetic hand functions with its design. The conventional prosthetic hands with robotic joins can achieve just 1 DOF per join [28,29]. Nevertheless, the proposed design has flexible joins, so that the small abduction/adduction on the joins will allow the fingers to easily adapt to the object forms. Basically, this improvement in the joins can provide multiple additional features to the prosthetic hand, such as hook, spherical grasp, cylindrical grasp, tip, etc. The proposed design also avoids for abduction/adduction limitations and permits a better prosthesis function, increasing the range of motion. The use of the force sensitive resistor placed on the distal phalange of the thumb allows one to define a better pressing over objects and stop the grasping of the prosthetic. Technical characteristics of the proposed prosthetic hand are presented in Table 4.

In Table 5, we presented the advantages and disadvantages of our prosthetic hand with respect to the existing ones.

The technique applied in the actual prosthetic hands are based on the fusion of the EMG and force sensitive technology. The combination of both techniques compensates the weakness of each of the techniques when operating independently.

Actual EMG prosthetic hands require some different movements done by the user to choose a specific movement manipulation as well as the pressure applied to the object surface [30]. This specific movement as multiple muscle contractions in a short time are sometimes not well accepted by the users [31]. To correctly control the force applied to the objects and the adequate fingers movement, the prosthetic hand requires the use of force sensitive sensors applied to all fingers individually. This method allows the prosthetic hand to correctly achieve the manipulation movement. In the proposed prosthetic hand, the EMG signals are used only to start the grasping. Nevertheless, the stop end of the grasp is controlled by the force-sensing resistors.

The sensory controlled prosthetic hand allows the user to actively enable the desired task. The sensor feedback gives to the user a more adequate control over the prosthetic hand and the external objects. The experimental results with the force sensitive resistors show good results on the grasping end, as well as imply minimum computational costs. When the pressure threshold is detected, then the grasping is stopped. Therefore, the grasping stop generates a new grasping order (flexion or extension), waiting for the new EMG signal and force sensitive resistor. When the new flexion (grasping) order is generated, the cycle is repeated again.

The integration of 5 force-sensing resistors will allow the actuators to work autonomously, i.e., with independent movement of the fingers. In this case, it will not be necessary to introduce muscle movement classifications for all types of grasping. This will decrease the EMG classification processing time and will reduce the memory space required for the signal processing. All these improvements will facilitate the real time functionality of the prosthetic hand, avoiding time delays. The sensor feedback allows the user to control the prosthetic hand and take decisions on the prosthetic hand activity. Vision also helps the user to define the prosthetic hand activity.

## 4. Conclusions

In the present work, the design and development of a prosthetic hand that emulates the human hand motion is presented. The proposed mechanical architecture of the prosthetic is based on the human hand anatomy and offers a broad range of movements. The articulation joins increase the degrees of freedom of the fingers and improve the hand flexibility, comparing with the existing bionic prosthetic hands mentioned in Table 5. The proposed prosthetic hand can perform fine movements and grasp different sized objects. The soft joins allow the prosthetic hand to achieve abduction and gyro on the fingers that adapt to the object surface. In the experiments with the tennis ball, the index and little fingers perform an abduction/adduction of 20–30 degrees. The weight of the device is 480 gr, which is below 500 gr and meets the requirements for bionic prosthetic hands. The system is externally powered and has robust and simple finger kinematics. The work shows that the employed materials must have enough flexibility and hardness to enable a correct use of the hand. The wire-driven tendons methodology that is employed in the proposed prosthetic hand solution shows a good grasping performance. The friction of the tendons is very low, which is another advantage of the prototype. The friction generated by the extensor tendons with respect to the tendon sheath, makes it that actuators with higher power are required.

On the other hand, the use of the EMG to control the prosthetic hand enables an increase in its ability when using human muscle actuation. The use of advanced signal processing (signal acquisition, filtering, classification, and training) is also remarkable to enable a correct simulation of the different hand movements. Finally, the usage of force-resistive sensors to end the grasping movement allows the prosthetic hand to simulate the touch pressure of the real hand.

In conclusion, the proposed solution shows interesting advantages versus available alternatives, enhancing the functionality and ergonomic nature of the device and not only relying on esthetic aspects.

## Figures and Tables

**Figure 1 sensors-21-00137-f001:**
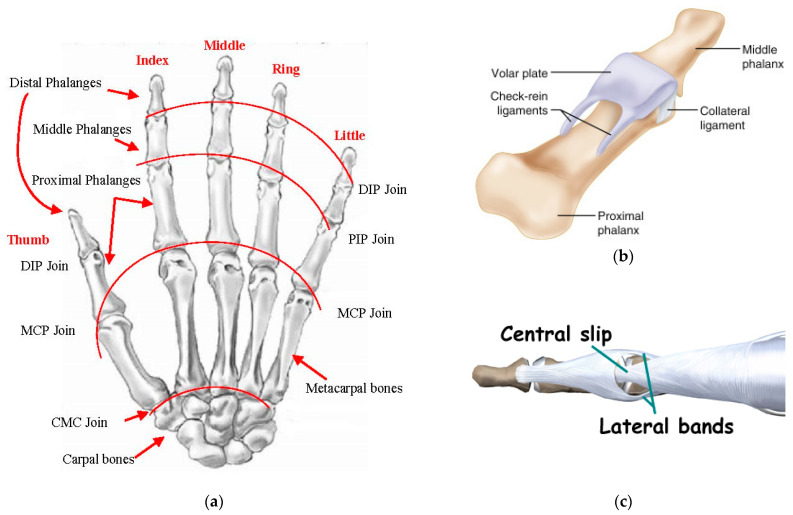
Human hand anatomy. (**a**) Human hand bones and joins; (**b**) volar plate and collateral ligaments; (**c**) extensor hooks.

**Figure 2 sensors-21-00137-f002:**
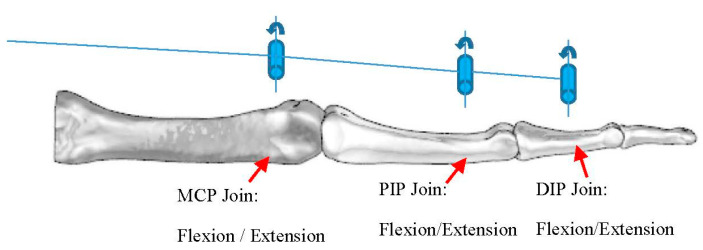
Three-dimensional model of the prosthetic hand assembly and its kinematics.

**Figure 3 sensors-21-00137-f003:**
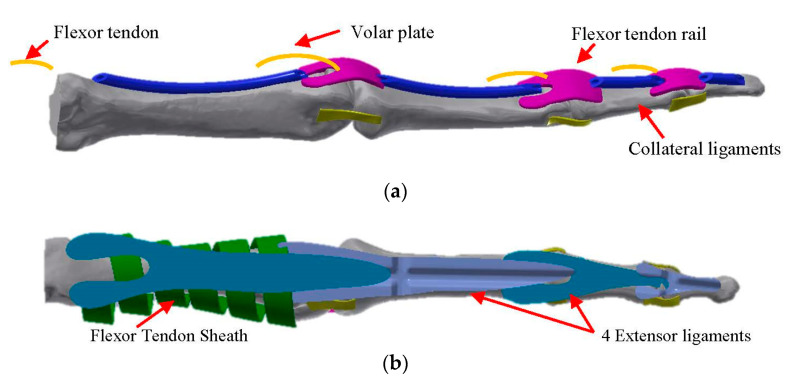
(**a**) Finger flexor tendon route and joins (two side collateral ligaments and volar plate). (**b**) Finger extensor tendons. Three-dimensional model of the prosthetic index finger assembly.

**Figure 4 sensors-21-00137-f004:**
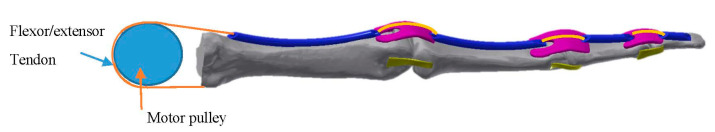
Mechanical architecture of the tendon transmission.

**Figure 5 sensors-21-00137-f005:**
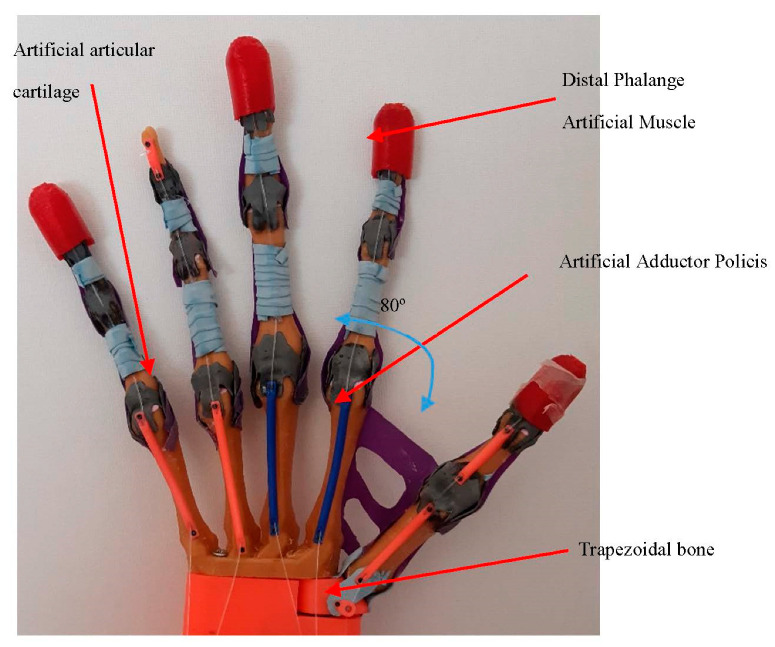
Mechanical assembly of the prosthetic hand.

**Figure 6 sensors-21-00137-f006:**
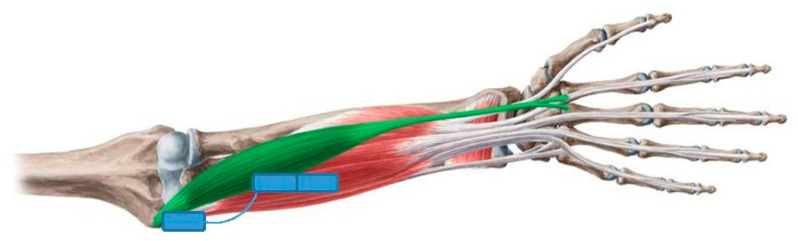
Positioning of the EMG sensors of the human hand.

**Figure 7 sensors-21-00137-f007:**
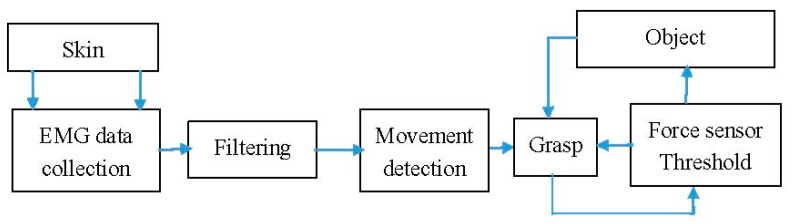
Motion sensing block diagram of the prosthetic hand.

**Figure 8 sensors-21-00137-f008:**
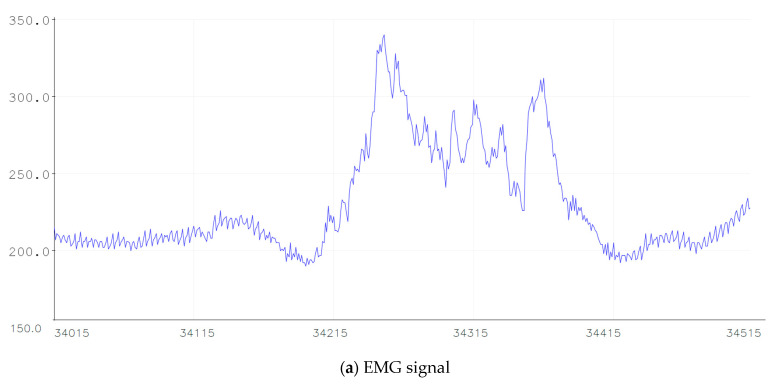
Example of EMG signal.

**Figure 9 sensors-21-00137-f009:**
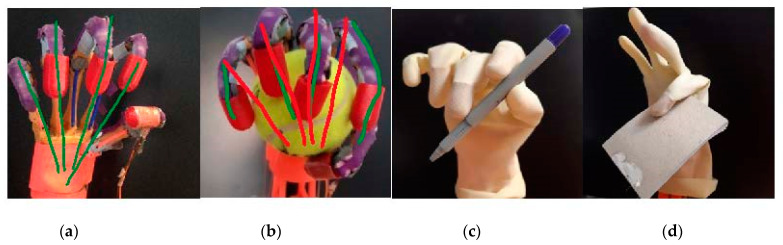
Prosthetic hand manipulation. (**a**) Grasping trajectory represented with green color, (**b**) grasp of a tennis ball without glove, (**c**) pencil grasp and (**d**) notebook grasp.

**Figure 10 sensors-21-00137-f010:**
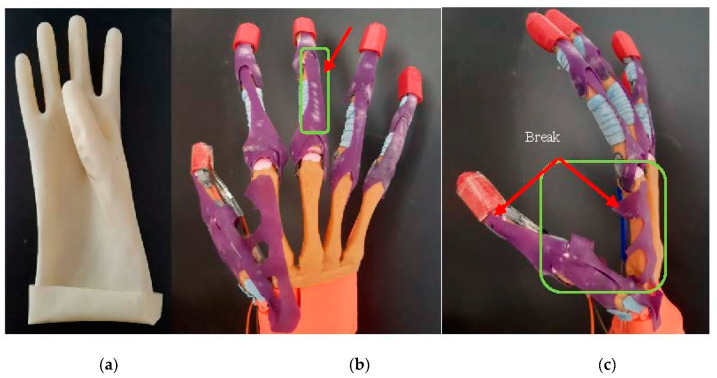
Rubber materials defects after 6 months of using. (**a**) The rubber glove form prevents the correct abduction of the thumb. The 90 degrees of the thumb placement does not allow the prosthetic hand to correctly manipulate the objects. (**b**) Extensor tendon wear due to friction with the tendon sheath. (**c**) Artificial Adductor policis (muscle) fracture.

**Table 1 sensors-21-00137-t001:** Bionic prosthetic hands characteristics.

	Model	Miquelangelo [22]	i-Limb [23]	Be Bionic [24]	Sensor Hand [25]	Vincent Hand [26]
Characteristic		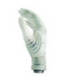	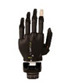	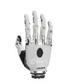	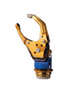	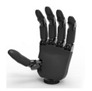
Developer	Otto Bock	Touch Bionics	Otto Bock	Otto-Bock	Vincent Systems
Weight, gr	510	599	500	500	410
Operating voltage, V	11.1	7.4	7.4		6–8
Battery type	li-ion	Lithium polymer	li-ion		li-Pol
Battery, mAh	1500	1300–2400	1300–2200		1300–2600
N° Actuators	2	6	5	1	6
Type of actuators		DC motor with worm gear	DC motor head screw	DC motor	DC motor worm gear
Active fingers	3	5	5	2	5
Force, N	70	100	140	100	60
Movement control	EMG, 4channels	Mobile app, EMG	SequentialEMG	EMG	Single triggerEMG
Movement command	Switching	Double and triple impulse	Co-contractions		Switch signalCo-contractionsDouble impulse
Feedback	NO	NO	Audible bipvibrations	NO	Vbrations

**Table 2 sensors-21-00137-t002:** Anatomic human hand dimensions.

Bone	Thumb	Index	Middle	Ring	Little
**Metacarpal Bone**	1.3567	2.049	1.906	1.719	1.578
**Proximal Phalange**	1.134	1.489	1.683	1.563	1.254
**Intermediate phalanges**		0.864	1	0.994	0.719
**Distal Phalange**	0.74	0.757	0.798	0.778	0.698

**Table 3 sensors-21-00137-t003:** Electronic components.

Components	Specifications
Arduino AtMega 2560	Input Voltage 7–12 V Analog Input Pins 16 DC Current per I/O Pin 40 mA DC Current for 3.3 V Pin 50 mA Clock speed 16 MHz EEPROM 4 KB SRAM 8 KB Flash memory 256 KB Analog inputs Pins 16 Digital Inputs 54
Myware EMG	Operating voltage 2.9 V–5.7 V Operating Current 9 mA–14 mA Output RAW and filtered signal
Force sensing resistors	Measuring range 0–2 kg Thickness <0.25 mm Precission ±2.5% Initial resistance >10 Mohm Voltage DC 3.3 V Response time 1 ms
Servo motor	Operating Voltage 4.8 V Operating current 50 mA Speed 0.12 at 4.8 V Torque 1.8 kg/cm Degree 180°
Step motor	Operating Voltage 5 V–12 V Operating current 2.5 A Speed 0.1 Torque 0.34 kg/cm
DRV8825 driver	Operating Voltage 8.2 V–45 V Operating current 2.5 A
Pushbutton Switch	12 mm button square

**Table 4 sensors-21-00137-t004:** Prosthetic hand characteristics.

**Weight, gr**	480
**Operating voltage, V**	12
**Battery type**	Li-ion
**Battery mAh**	3000
**N actuators**	6
**Type of actuators**	DC motor
**Active fingers**	5
**Force, N**	17
**Movement control**	EMG, FSR
**Movement command**	Buttons, co-contractions
**Max flexion degree per join, °**	90
**Max abduction per join, °**	30
**Max Thumb abduction, °**	90
**Joins**	soft
**Feedback**	NO

**Table 5 sensors-21-00137-t005:** Prosthetic hand advantages and disadvantages.

	Advantages	Disadvantages
**Miquelangelo**	- Better grasping wit multifunctional fingers - Wrist rotation - Multiple grip types	- Difficulties on co-contractions too much effort - External device requires as mobile phone, buttons, etc. - No need of gyroscope - Pattern recognition requires much more training - Expensive - Expensive repair - Heavy objects the fingers open unintentionally - Insufficient force - Grasp of static objects - Mechanical robustness
**i-Limb**		
**Be bionic**		
**Sensor Hand**		
**Vincent Hand**		
**Our Hand**	- Soft joins - Adaptation to the object surface - Multiple grip types - Combination of 2 control systems EMG and FSR	- Difficulties on co-contractions and or bottom (needs the other hand) - Insufficient force - Mechanical robustness

## Data Availability

Data available on request due to restrictions. The data presented in this study are available on request from the corresponding author. The data are not publicly available.

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
