# Peer review of "Human Hand Anatomy-Based Prosthetic Hand"

_sensors, 2020, doi:10.3390/s21010137_

Round 1

Reviewer 1 Report

The present paper describes the development of a prosthetic hand based on the human hand anatomy. The grasping task is started using surface EMG sensors. The paper is interesting and easy to read. However, the following points need to be clarified before publication. 

The experimental setup and the experimental protocol should be better described and commented. In particular, it should be mentioned how many subjects tested the prototype, what is their age, gender and physical conditions. 

Since experiments with human subjects have been performed, it is not clear if the experimental protocol was approved by an Ethics Committee, and if participants gave informed consent prior to start the experiments.

It would be interesting to introduce some metrics in order to better evaluate the performance of the prototype. A brief comment about this point would be appreciated.

The text should be checked again in order to match the template rules. Furthermore, some details are missing in References 17, 18, and 19.

Author Response

The present paper describes the development of a prosthetic hand based on the human hand anatomy. The grasping task is started using surface EMG sensors. The paper is interesting and easy to read. However, the following points need to be clarified before publication. 

The experimental setup and the experimental protocol should be better described and commented. In particular, it should be mentioned how many subjects tested the prototype, what is their age, gender and physical conditions. 

We thank the reviewer for his kind comments. Following his suggestions, additional information concerning the experiments has been added in Section 3. Also, an additional figure has been included.

Since experiments with human subjects have been performed, it is not clear if the experimental protocol was approved by an Ethics Committee, and if participants gave informed consent prior to start the experiments.

In this work stage, the prototype was tested only with the authors of the work since the objective of the proposed research is to define prosthetic hand ability, design versus functionality. The next step is to improve the prosthetic hand and test it massively with human volunteers in the controlled environment by the traumatologist and specialists.

It would be interesting to introduce some metrics in order to better evaluate the performance of the prototype. A brief comment about this point would be appreciated.

We include some figures related to the movement features of the prototype.

The text should be checked again in order to match the template rules. Furthermore, some details are missing in References 17, 18, and 19.

Thanks for the note. We corrected the stated references.

[17] be bionic, Technical Manual, Available online https://shop.ottobock.us/media/pdf/bebionicTechManualSmall.pdf, (accessed on August 2020). 

[18] i-limb, Available online:  https://www.ortosur.es/catalogo-de-productos/protesis/miembro-superior/mano-mioelectrica/i-limb/, (accessed on 27 October 2020).

[19] Dosen S., Cipriani Ch., Kostic M., Controzzi M., Carrozza M.C., Popovic D.B., Cognitive vision system for control of dexterous prosthetic hands: Experimental evaluation. J. Neuroeng. Rehabil., 2010, 7-42.

Reviewer 2 Report

The paper presents aa prosthetic hand based on the human hand anatomy. The paper deals with an interesting topic. The study itself is generally understandable and well written. The study structure is correctly defined and the introduction is suitable to present the problem that the Authors aim to face. In order to increase the impact of the proposed research, I suggest to improve the contribution of their proposal respect to the papers already available in literature (citing the related references) on the topic. In particular, it may be useful for the readers to develop a table that present and compare similar devices underlining advantages and limitations that their device is able to solve. This point may help to highlight the novelty of the proposed study. The device is well described and the main idea is justified and supported by figures. The application of force sensitive resistors for simulating the hand touch pressure increases the capability of the device in simulating the human movement. However, the presented results are related to simulation of a potential application, proving the effectiveness of the presented devices. The experimental tests are not correctly described, what are the main quantitative indicators to evaluate the effectiveness of the device? The paper shows only a set of figures and the description of the tests is very poor. Please, provide how the test are executed and evaluated. Finally, my suggestion is to explain in more details how the device may be installed and integrated to solve problems of the common usage of a patient. The conclusions are poor respect to the potentialities of the device.  

Minor comments:

  1. Improve the quality of Figure 7
  2. Figure 10 needs to be better described

Reviewer 3 Report

The topic is interesting, and probably the work done by the authors is useful to the scientific community, but:
1) the presentation is confused, it is very difficult to understand its contents;
2) it is not clear what the novelty would be compared to the state of the art, given that each finger has only one DOF, although with the possibility of abduction / adduction, thanks to the elasticity of the materials;
3) authors talk about EMG features, but are not described, if not an amplitude threshold, as well as the classification of the features;
4) it is not clear what is the usefulness of EMG signals with respect to force sensing resistors to control grasping;
5) there are obscure phrases, such as "the whole load is based on the actuator axis" (246), or "The threshold is defined in accordance with the EMG signal voltage for each type of grasping" (199-200);
In practice I would say that the novelty of the presented work consists mainly in the materials used, which allow the prosthetic hand to be more flexible.

Author Response

We have added an additional table in the introduction section to synthesize the main features of other works explained during that section.

2) it is not clear what the novelty would be compared to the state of the art, given that each finger has only one DOF, although with the possibility of abduction / adduction, thanks to the elasticity of the materials;

The device has 15 DOF (Degree of Freedom) one DOF per each join. The soft material joins provide prosthetic hand a high level of adaptation to the object surface. They increase DOF of each join, enabling the small abduction/adduction added to the flexion / extension.

3) authors talk about EMG features, but are not described, if not an amplitude threshold, as well as the classification of the features;

The features of the motion movement are not implemented in the present experiment.

4) it is not clear what is the usefulness of EMG signals with respect to force sensing resistors to control grasping;

As mentioned in [25] more than 45% of the users reject the usage of the biosensing prosthetic hand. The main motive is the lack of feedback and sensing control. The technique applied in the actual prosthetic hand is based on the fusion of the EMG and force sensitive technology. The combination of both techniques compensates the weakness of each of the techniques when operate independently. The EMG control system does not provide feedback to the user as well as the force applied to the objects is not controlled. In order to control the hand movements, it requires a complex pattern recognition algorithms of the EMG signal as well as user interaction.

Actual EMG prosthetic hands requires some different movements done by the user in order to choose a specific movement manipulation as well as the pressure applied to the object surface [y]. This specific movement as multiple muscle contractions in a short time in some times are not well accepted by the users [z]. To correctly control the force applied to the objects and the adequate fingers movement, the prosthetic hand requires the use of force sensitive sensors applied to all fingers individually. This method allows the prosthetic hand to correctly realize the manipulation movement. In the proposed prosthetic hand, the EMG signals are used only to start the grasping. Nevertheless, the stop end of the grasp is controlled by the force sensing resistors.

5) there are obscure phrases, such as "the whole load is based on the actuator axis" (246),

We have tried to modify the unclear sentences as that one pointed out by the reviewer.

or "The threshold is defined in accordance with the EMG signal voltage for each type of grasping" (199-200);

The threshold is defined in accordance with the EMG signal voltage. To this end, some experiments were developed to classify different grasp types but they were not included in the paper due to page limitation.

In practice I would say that the novelty of the presented work consists mainly in the materials used, which allow the prosthetic hand to be more flexible.

Thanks a lot for your comments.

Round 2

Reviewer 3 Report

The authors' review improved the paper, explaining the methodology better and making reading easier. Although the paper presents only preliminary results, it offers a useful contribution to researchers in this area, pending further study.